# Qualitative description of interpersonal HIV stigma and motivations for HIV testing among gays, bisexuals, and men who have sex with men in Ghana's slums—BSGH-005

Gamji Rabiu Abu-Ba'are[1,2,3,4,5,6], Osman Wumpini Shamrock[1,2]*, Edem Yaw Zigah[1,2], Adedotun Ogunbajo[7], Henry Delali Dakpui[1,2], George Rudolph Kofi Agbemedu[1,2], Donte T. Boyd[8], Oliver C. Ezechi[5], LaRon E. Nelson[4,9], Kwasi Torpey[6]

1 Behavioral, Sexual and Global Health Lab, School of Nursing, University of Rochester, Rochester, New York, United States of America, 2 Behavioral, Sexual and Global Health Lab, Jama'a Action, West Legon, Accra, Ghana, 3 Department of Public Health Sciences, University of Rochester, Rochester, New York, United States of America, 4 School of Nursing, Yale University, New Haven, Connecticut, United States of America, 5 Clinical Sciences Department, Nigerian Institute of Medical Research, Lagos, Nigeria, 6 Department of Population, Family and Reproductive Health, School of Public Health, University of Ghana, Accra, Ghana, 7 Us Helping Us People into Living, Washington, DC, United States of America, 8 College of Social Work, Ohio State University, Columbus, Ohio, United States of America, 9 Center for Interdisciplinary Research on AIDS, School of Public Health, Yale University, New Haven, Connecticut, United States of America

* osmanwumpini_shamrock@urmc.rochester.edu

## Abstract

Despite significant progress in Ghana's HIV response, disparities in HIV prevalence persist among different populations. Gays, bisexuals, and other men who have sex with men (GBMSM) in the country remain vulnerable to HIV infection due to high levels of stigma and discrimination, limited access to healthcare services, and low HIV knowledge levels. While limited studies focus on HIV prevention and care in the Ghanaian GBMSM context, we did not find studies on GBMSM in slums. We, therefore, explored stigma and motivations of HIV testing among GBMSM in slums. In collaboration with our community partners, we recruited and conducted face-to-face interviews among 12 GBMSM from slums in Accra and Kumasi, Ghana. Our multiple-reviewer summative content analysis identified the following: under HIV stigma, we identified two categories, avoidance of GBMSM living with HIV and fear of testing positive for HIV. Under motivations for HIV testing, we identified three categories; HIV vulnerability, knowing one's HIV status, and positive messaging about HIV. Our findings provide valuable insights into stigma and motivations for HIV testing among GBMSM in Ghanaian slums. They also highlight the importance of targeted HIV education interventions to empower GBMSM to take responsibility for their sexual health and address the unique challenges they face accessing HIV testing services.

**Data Availability Statement:** All data contain in the study can be found in the manuscript.

**Funding:** GRA applied and received grant funding from the Yale university FLAGS grant. Funding agency did not play a role in the study design, data collection, and analysis, decision to publish or preparation of the manuscript. All content is solely those of the authors and does not represent that of the funding agency.

**Competing interests:** The authors have declared that no competing interests exist.

## Introduction

Sub-Saharan Africa (SSA) carries over two-thirds of the world's burden of HIV, yet, HIV testing and related services remain underutilized due to several factors such as insufficient knowledge, low-risk perception, and increased stigma [1,2]. Despite programmatic efforts in SSA, the burden of HIV and the factors (e.g., stigma) that hinder access to HIV testing remain a significant obstacle to HIV testing among HIV key populations such as gays, bisexuals, and men who have sex with men (GBMSM) [3–6].

Although Ghana has made significant strides in its HIV response, disparities persist in HIV prevalence between different populations, with GBMSM carrying a disproportionate burden, 18% compared to that of the general population, 1.7%% [7–9]. Efforts to increase HIV testing and prevention among GBMSM continue, yet reports indicate suboptimal testing rates in this group in the country due to individual, interpersonal, and environmental barriers [6–8,10,11]. At the individual level, such factors include low-risk perception, low HIV knowledge, and fear of rejection [3,12]. At the environmental level, factors include healthcare facility-level stigma, inadequate access to testing, and community stigma among others [12–15]. At the peer level, interpersonal HIV stigma affects interest in testing as some GBMSM discriminate against their peers living with HIV [3,12].

While limited studies explore stigma and testing practices among GBMSM in Ghana, no known studies have focused explicitly on GBMSM in Ghanaian slum communities. However, in SSA, studies have highlighted the association of low HIV knowledge and low HIV testing to persons with low socioeconomic status [16,17]. Slums also remain associated with high-risk behaviors such as transactional sex, inconsistent condom use, and increased HIV prevalence [18–20]. Emerging findings as from 2010 in eastern Africa show that GBMSM in slums have low-risk perception, increased risk behaviors, and low access to HIV testing and prevention services [21,22]. In 2012, HIV rates among slum residents in Kenya were notably higher at 12%, compared to 5% among non-slum urban residents and 6% among rural residents [22]. Also, out of 4028 youth sampled in Kenyan slums, only 27% had ever tested for HIV, and over 90% had low HIV risk perception. Of those who tested, over 90% reported being required to take the test [21].

Whereas no specific studies exist among GBMSM in Ghana's slums, previous studies among slum communities in Ghana report increased HIV risk behaviors and higher HIV prevalence among slum residents [19,23,24]. Communities with increased HIV prevalence and risk also have poorly resourced infrastructure and health facilities which contributes to poor health outcomes [25,26]. Urban areas such as the Accra and Kumasi larger regions present the highest prevalence rates of HIV, 2.47% and 1.98%, respectively, compared to the other 16 regions [9]. These two cities and their surrounding areas also record high prevalence among GBMSM, Accra (42%) and Kumasi (25%), compared to the national GBMSM rate of 17.5% [7]. Thus, placing GBMSM living in urban slums at increased risk of HIV infection than other populations.

The current study seeks to understand GBMSM slum-specific HIV stigma manifestation and motivation for testing in Accra and Kumasi, Ghana. Understanding stigma and motivations for HIV testing is critical to reducing HIV transmission and improving health outcomes for this population.

### Self Determination Theory (SDT)

The Self Determination Theory (SDT) explains, individuals' innate psychological needs inform their wellbeing and quality of life within a social context [27,28]. The various components of SDT (the basic psychological need for autonomy, competence, and relatedness) could help

explain the importance of HIV testing motivations among GBMSM in slums. Per the psychological need for autonomy, the barriers faced by GBMSM in slums can undermine the autonomy and restrict their ability to make informed HIV testing [12,13,29–31]. On the psychological need for competence, the low levels of HIV knowledge can affect GBMSM's proficiency to make informed decisions regarding HIV testing [29,32].

The theory also offers insight into HIV stigma by GBMSM. Being stigmatized can affect the autonomy GBMSM have over their ability to test for HIV due to the potential threat of disclosure of their sexual orientation without their consent. This perceived threat may lead to the reluctance of GBMSM to seek HIV testing. GBMSM may be affected by negative stereotypes and misconceptions about homosexuals and HIV. These misconceptions may lead to situations where GBMSM feel ashamed, have self-doubts, or have low self-esteem, affecting their chances of testing for HIV. The stigma and discrimination faced by GBMSM can also negatively impact the relatedness aspect of SDT, which emphasizes the need for individuals to feel connected and supported in their social environments. Creating supportive and inclusive environments, reducing stigma, and improving access to healthcare services are crucial for promoting relatedness and encouraging GBMSM to seek HIV testing [33–35].

## Methodology

### Study design

The study used a qualitative interview [36] to collect data from GBMSM living in slum communities in Ghana. This approach allowed researcher in this study gather firsthand experiences of GBMSM around HIV stigma and motivations for HIV testing for those living in slum communities within the Accra, and Kumasi cities in Ghana.

### Sampling and recruitment procedure

Using the time location sampling (TLs) technique, we reached and recruited GBMSM in slum communities in collaboration with our community partners in Accra and Kumasi. Research assistants working with our community partner organizations in Accra. community partner organizations screened and invited GBMSM to take part in interviews sessions during one of the organizations' activities when GBMSM visited the site. We have worked with these community partners in previous studies [10,11]. Although we originally intended to purposively include 19 participants in this study, we reached saturation of information after the eight interviews. When responses from study participants were found to be consistent by the research assistants, an additional four transcripts were included to ensure complete information saturation, bringing the total number of transcripts to twelve.

### Study setting

Ghana is a country located in West Africa and the main religions practiced are Christianity (71%), Islam (18%), and African Traditional Religion (5%) [37]. The education statistics show that the majority of children (71%) complete primary education, but the percentages decrease (47%) for lower secondary school, and upper secondary school (35%) [38]. A study conducted on MSM (transgender, gay, bisexual, and straight) in Ghana revealed that over half (52.2%) of them are educated, with around 16.8% of them having tertiary or higher educational qualifications [7]. Additionally, the study highlighted that nearly half (44.1%) of the MSM were employed and had never been married [7]. The official language spoken in Ghana is English, but there are several indigenous languages such as *Twi*, *Ewe*, and *Fante* that are spoken more frequently [39].

## Inclusion criteria

Participants in the study had to be at least 18 years old and live in a slum community in Accra, Ghana's Greater Accra regional capital, or Kumasi, Ghana's Ashanti regional capital. Additionally, the individual identity had to be a cisgender man who self-identifies as gay, bisexual, or pansexual or engage in sexual intercourse with another cis-gender man for reasons other than sexual orientation. The person must have been sexually active during engagement and must have had sexual intercourse with another cis-gender man within the previous six months.

## Data collection procedure

**Procedure.** To gather information from the participants, we conducted in-depth face-to-face interviews. Following the screening, participants were given consent forms by the research assistants to review. Research assistants (EYZ, OWS) engaged in data collection had prior and extensive research training from working with these populations in the past. We also utilized the expertise of these researchers in our past studies, with similar populations, hence, the familiarity and experience engaging our participants [23,24,40–44]. The research assistants also read the consent forms out loud and provided extra explanations to ensure everything was understood. They answered the participants' queries before the interviews started, collected signatures confirming GBMSM's agreement to participate in the study, and allow for audio recording. The community partners' private spaces were used for all interviews. All but four interviews were in English, the other four in *Twi*, a local Ghanaian language that some participants found more conversant. Data collected in *Twi* were translated and transcribed into the English Language. Data were collected from participants in January 2022, and lasted for 10 days.

**Nature of questions.** The research assistants were trained to conduct qualitative interviews using the study's checklist as a guide in collecting information on HIV stigma and motivations for testing among GBMSM living in Ghanaian slum communities. In line with our design, the checklist allowed a more transparent and open discussion rather than the traditional question-and-answer interview structure. Participants were asked to share their experiences of HIV testing, their knowledge, and what motivated or affected their interest in testing.

**Analytical strategy.** Trained research assistants deidentified the transcripts after translating the audio interview recordings verbatim. We performed a summative content analysis on the transcripts with multiple reviewers [11]. Our team has successfully used this analytical method to comprehend crucial components in participant accounts [11]. Each transcript received at least two reviewers. Each reviewer independently read the transcripts to identify the most statements made by the participants. They then reported these statements in between 100 and 200 words. The principal authors reviewed each summary to find clusters and recorded the elements frequently appearing in transcripts and summaries in a data spreadsheet. We identified several clusters and classified them under categories that outlined participant experiences, perceptions, and motivations for HIV testing. Each area that was reported appeared in both peer reviewers' summaries.

**Ethical considerations.** Ethical approval was received from the Ghana Health Service Ethics Committee (GHS-ERC 011/10/21) and the Institutional Review Board Committee (IRES IRB) of Yale University (IRES IRB #RNI00002010). The interviewers in this study ensured that each participant had read and understood the informed consent form thoroughly before any data was collected, and afterward, they obtained written consent.

## Findings

### Description of participants

The 12 participants identified as cis-gender men and had sexual intercourse within the previous six months with another cis-gender man. Six participants identified as Christians, four as Muslims, and two as both Muslims and Christians. Five participants accomplished tertiary education, and six concluded senior high school while one didn't complete Junior High school education.

### Description of findings

Findings from the data collected and analyzed revealed two categories; 1) HIV stigma and 2) motivations for HIV testing. Under HIV stigma: avoidance of GBMSM living with HIV, and fear of testing positive for HIV. Under the category motivations for HIV testing: HIV vulnerability, sexual health decision-making, and knowing one's HIV status.

### HIV stigma

**Avoidance of GBMSM living with HIV.**   Participants in the study indicated they were afraid and had negative feelings about people living with HIV. One person, Participant K, mentioned they were scared to be near someone because of fear of a potential HIV infection. According to the participants, this fear of HIV stemmed from simple things like eating, sleeping in the same room, and bathing with persons living with HIV. Others, such as Participant F, thought that only people who have a lot of sex are likely to get HIV. They believed that married people were less likely to get the virus because they weren't promiscuous but had sex with only their marriage partners. However, not everyone felt this way. Some participants were well informed about how HIV is transmitted and were comfortable with being around people who have HIV. Participant E, for example, said they would do things like eat or hang out with someone who has HIV, but they wouldn't have sex with persons living with HIV.

> I'm scared of them. I will be scared to eat with the person infected with HIV. Bath with the person or even live in the same area with the person. Because I don't want to get close to the person, maybe because my thoughts are that the person may transfer it to me at any moment, so I will neglect the person (GBMSM participant K).

> They do much of sex, they are sex addicts. Because when you don't have sex, you won't get infected. And married people usually don't get infected in my community (GBMSM participant F).

> I wouldn't mind eating with the person because I was told you can only get infected through sex, deep kissing and sharing sharps. So I will treat the person like an ordinary person. Sex will be the only thing I wouldn't have with this person. I am nondiscriminatory; that's why my friends came to me and told me they were positive (GBMSM participant E).

**Fear of testing positive for HIV.**   Participants mentioned they were really scared about getting tested for HIV. One person, Participant B, felt nervous about telling others their HIV status and was scared at the idea of getting tested. They worried that they might test positive for HIV. Another person, Participant E, said getting tested for HIV was scary at first because they were afraid of testing positive and what would happen next. However, both participants talked about how they learned to deal with their fears. Participant B said it was important to encourage themselves and think positively about the testing process. Participant E said they

used to find HIV testing scary, but now they see it as a regular thing, like other medical tests. The participants indicated the importance of clearing up any wrong ideas and stigma about HIV testing so people won't be so scared and will be more open to getting tested.

> It's scary. I have tested for HIV, but I wouldn't like to share my status now. Being in the house, I encourage myself to go and get tested. However, when I get there, it's scary. What if it comes out positive? But rather, I should be thinking the other way that it should be negative (GBMSM Participant B).

> It's very scary going for HIV testing, but when you encourage yourself to do the test, you realize it's not that scary but just like the normal tests that we do. But because of the mentality that if you test, you will come out positive and the ARV and other stuff, its scary (GBMSM Participant E).

## Motivations for HIV testing among GBMSM

**HIV vulnerability.**    Some GBMSM were motivated to test for HIV due to their HIV high-risk awareness, as being a sexually active GBMSM poses a higher risk of getting HIV because of the increased HIV prevalence compared to the general population. Participant A mentioned it was really important for MSM to get tested for HIV for these reasons. Even though they were scared, some participants understood that it was imperative to know if they had HIV to stay healthy. According to Participant D, they were scared to get tested, but they knew it was necessary to prevent getting sick. They were worried about having past unprotected sex and wanted to make sure they were safe by getting tested for HIV. Other participants indicated how they regularly get tested for HIV to stay healthy. Participant C said they never know what might happen during sex and they could catch HIV without knowing it. They wanted to be careful and make sure they stayed healthy, so they made sure to get tested regularly.

> Yes, there's the need to test for HIV. . .because men who have sex with men have a higher rate of getting HIV infections compared to men having sex with ladies. So, it's very good for any MSM to get tested (GBMSM participant A).

> It is very scary when I am going to test. But I feel it's very important for me to know my status to prevent me from getting sick. So, it's very important for me to test for HIV. I test for HIV sometimes because maybe I've had raw sex with someone, and I don't trust the person (GBMSM participant D).

> I do test for HIV because it's an opportunistic infection. And you will not know when you might contract it, so I have to test for it to know my status. . .the sex doesn't always go as planned. . .and risky behaviors. I have had some infections some time ago. And now I prefer to be extra careful (GBMSM participant C).

**Knowing one's HIV status.**    GBMSM participants shared their thoughts about why HIV testing was important, pointing out different reasons related to their health and wellbeing. According to Participant Z, getting tested for HIV is important for knowing one's status and making informed decisions. They were assured that medications are available to treat HIV and allow one to live a long and healthy life if diagnosed early. Participant F emphasized the urgency of HIV testing to prevent complications and the progression to AIDS. They stressed the risks of delayed testing and the importance of acting quickly to avoid negative health

outcomes. Similarly, Participant I mentioned the importance of timely HIV testing after potential exposure to HIV. They described feeling uneasy after sexual encounters and felt it was necessary to get tested promptly to address any risks.

> Of course, it is necessary to get tested because it will help you to know your status and decision-making. If I test positive for HIV, there are medications to help me live long (GBMSM participants Z).

> It's important for me to test for HIV...because you have to know your status. Knowing your status allows you to know what to do. The more you keep on delaying, the more the thing (HIV) too gets worse. Because maybe you are HIV positive, you aren't on drugs, and if you delay, it might turn into something else (AIDS). And that can cause your death (GBMSM participant F).

> I feel it's important to test for HIV because sometimes you feel like you've been exposed (to HIV). Sometimes after having sex, you feel as if you should go, and your body isn't feeling too well. These thoughts just run through my head, so I go and test because I'm not comfortable with that. So, the earlier you get tested, the better (GBMSM participant I).

**Positive messaging about HIV.** Several participants shared their experiences and views on HIV testing, revealing a range of motivations and beliefs related to health. Participant O took a proactive approach to HIV testing, stressing the importance of knowing one's status in light of growing awareness about HIV transmission. They described their decision to get tested and their relief at receiving a negative result. Participant O also showed knowledge of HIV treatment options, highlighting the potential for achieving an undetectable viral load with proper medication adherence and emphasizing the overall benefit of HIV testing for health management.

Participant D also talked about their changing understanding of HIV and the significance of everyone getting tested. They explained how learning about HIV shifted their perceptions of its severity compared to other diseases, like malaria. They also emphasized the right to life for people living with HIV and spoke out against stigma and discrimination. Participant C reechoed the idea of proactive testing and understanding treatment options. They mentioned the reassurance of treatment availability as a reason for regularly getting tested.

> Lately, I have been hearing a lot about HIV and how it's spreading. I went and took the test. And I found out I'm negative. It is always good to know your status. When I was admitted to college, I had my test before I started lectures. Even if you test positive there are treatment options for you and I learned if you take your medication well, you will achieve the undetectable stage. So, I believe is good to test for HIV (GBMSM Participant O).

> I thought HIV was so deadly until I started reading about it. I learned that malaria is even more dangerous than HIV, so I think everyone should test and know their status. And I believe persons living with HIV have the right to life, and they shouldn't be stigmatized or discriminated against... I've learned that there are drugs they give to them when they have HIV. And when they give it to them and take it as prescribed, you will be ok and fine (GBMSM participant D).

> I was told that if I should test positive for HIV, there were treatment options and other medications which would prolong my life. So, I always test when it's time for me to test (GBMSM participant C).

## Discussion

Despite the heightened stigma, elevated vulnerability to HIV, and low rates of HIV testing among GBMSM and residents of slums, there is a lack of research exploring HIV testing among GBMSM, particularly within slum communities in Ghana [11,45–47]. The SDT theory was useful in the context of this study to highlight the psychological needs for autonomy, competence, and relatedness for GBMSM in HIV testing, or how this affects the motivation for testing. The theory sheds light on the barriers and facilitators of HIV testing that' could affect GBMSMS living in the slum communities. The theory also improved our understanding of HIV stigma and how it impacts autonomy considering that the fear of negative stereotypes impedes uptake of testing. GBMSM stigma around HIV testing may affect testing knowledge or proficiencies. Stigma also affects relatedness, which is a crucial component of social support.

The present study qualitatively describes GBMSM level stigma and their motivations for HIV testing in Ghana's slums. Whereas some GBMSM demonstrated an understanding and acceptance of people living with HIV, others avoided their peers living with HIV and had fears of testing for HIV. The need to test was driven by factors such as perceived HIV vulnerability, need to know one's HIV status for sexual health reasons, and positive messaging about HIV-informed motivations for HIV testing. These findings emphasize the necessity for interventions that enhance HIV awareness and capitalize on existing motivation to enhance HIV testing rates among GBMSM residing in slum areas [4,48–51].

Although this is one of the early studies to explore GBMSM level HIV stigma in slum communities, the findings remain consistent with literature reported among GBMSM and other populations in Ghana and other SSA [4,48–53]. Consistent with previous findings and aligned with key components of autonomy and competence in the SDT, HIV stigma undermines HIV testing decisions among GBMSM as they fear testing for HIV due to the stigma associated with testing positive [10,52–54]. Thus, it remains imperative for interventions to address HIV stigma to improve testing [10,54,55]. The present study findings of participants projecting stigma towards their peers living with HIV by avoiding them, refusing them sex, and labelling them as sexually promiscuous was also reflected in our previous findings among other GBMSM. Such labeling and avoidance of others living with HIV reflect our previous findings among other GBMSM in Ghana [10,30,48,54,56]. Such behavior poses a significant challenge for testing and the willingness of GBMSM who test positive to disclose their status and even adhere to care as they will not want to be isolated by their peers [56–58]. This finding is also imperative in understanding the relatedness component of the SDT framework that explains HIV stigma from peers may lead to avoidance or social isolation leading to the reluctance to seek social support services.

Despite the stigma shown by others, findings around motivations for HIV testing provide opportunities for encouraging GBMSM in slums to improve HIV knowledge and testing among their peers [11,12,45]. As shown in the results, some GBMSM understand the basics of HIV, especially around risk behaviors, and respond to such risk by testing for HIV, which remains consistent with previous research that shows that the awareness of increased vulnerability among GBMSM encouraged them to test [13,48,59]. Consistent with the SDT and prior literature, the participants' responses show that increased knowledge of HIV will increase their self-determination for testing. GBMSM did not only consider HIV testing as a path to knowing their HIV status but as an essential means for knowledge acquisition about their health to enable them take meaningful steps like seeking care and adopting behavior changes to ensure their wellbeing. Whereas no previous studies focused on GBMSM in slums, earlier studies among GBMSM in Ghana showed that some GBMSM acknowledged the importance of HIV

testing, as it enabled them know their status and make important sexual health choices [13,45,48].

Providing information on how HIV is transmitted, participants' accounts show that positive messaging about HIV, instead of negative messaging can encourage HIV testing. Others informing participants about how HIV is transmitted and treated, including stages such as undetectable status motivated them to test for HIV. Previous studies reported similar findings among GBMSM [10,52,54]. In one of the studies on HIV health promotion, after attending a workshop where GBMSM peer groups discussed and learned about HIV, they observed an increase in HIV testing from 4% to 17% within a week post-intervention [10].

Taken together, the central takeaway from the findings on stigma and motivation for HIV testing lies in the importance of empowering GBMSM to take charge of their sexual health and make informed decisions about HIV testing, as this may help in reducing barriers to testing and promote self-efficacy. By enhancing their sense of competence and autonomy, tailored interventions can motivate GBMSM living in Ghanaian slums to overcome barriers to HIV testing and take control of their sexual health. A popular intervention that could help reduce HIV stigma at the individual and interpersonal level is the Many Men Many Voices (3MV) [10,54,60]. The intervention addresses stigma, HIV risk, transmission, testing, and treatment among GBMSM in Black communities. When adapted to Ghana, we found that GBMSM improved their understanding of HIV, formed community, and improved their HIV testing behaviors [54].

## Conclusion

Despite the important findings, research needs to consider the study limitations when interpreting these study findings. As a qualitative study, which recruited from two regions in Ghana, the findings may not apply to all GBMSM in slums across Ghana. We therefore recommend using this study's findings in conjunction with other studies from Ghana to draw conclusions about HIV stigma and testing among GBMSM. Future studies could include GBMSM from other regions and consider targeting specific age groups, as this formative work did not have any specific age brackets and may not fully represent people in different age groups.

In conclusion, our findings contribute to the existing knowledge and provide insights for policymakers, healthcare providers, and researchers to develop effective strategies and programs aimed at reducing HIV disparities and improving HIV testing among GBMSM in slums. While stigma can undermine HIV testing, some GBMSM are highly motivated to test for HIV as such positive messaging about HIV should be encouraged and leveraged to increase HIV self-testing among GBMSM in Ghana's slums.

## Author Contributions

**Conceptualization:** Gamji Rabiu Abu-Ba'are, Osman Wumpini Shamrock, Edem Yaw Zigah, Henry Delali Dakpui, George Rudolph Kofi Agbemedu.

**Formal analysis:** Gamji Rabiu Abu-Ba'are, Donte T. Boyd.

**Funding acquisition:** Gamji Rabiu Abu-Ba'are.

**Investigation:** Edem Yaw Zigah, George Rudolph Kofi Agbemedu, LaRon E. Nelson.

**Methodology:** Gamji Rabiu Abu-Ba'are, Osman Wumpini Shamrock, Adedotun Ogunbajo, Henry Delali Dakpui, George Rudolph Kofi Agbemedu, Oliver C. Ezechi.

**Supervision:** Kwasi Torpey.

**Visualization:** Gamji Rabiu Abu-Ba'are, Osman Wumpini Shamrock, Henry Delali Dakpui, George Rudolph Kofi Agbemedu, Oliver C. Ezechi, LaRon E. Nelson.

**Writing – original draft:** Gamji Rabiu Abu-Ba'are, Osman Wumpini Shamrock, Edem Yaw Zigah, Adedotun Ogunbajo, Henry Delali Dakpui, George Rudolph Kofi Agbemedu, Donte T. Boyd, Oliver C. Ezechi, LaRon E. Nelson, Kwasi Torpey.

**Writing – review & editing:** Gamji Rabiu Abu-Ba'are, Osman Wumpini Shamrock, Edem Yaw Zigah, Adedotun Ogunbajo, Donte T. Boyd, Oliver C. Ezechi, LaRon E. Nelson, Kwasi Torpey.

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
