## [Decision Letter · Decision Letter 0]

4 Apr 2024

PONE-D-23-23534Qualitative description of interpersonal HIV stigma and motivations for HIV testing among gays, bisexuals, and men who have men in Ghana's slums- BSGH-005PLOS ONE

Dear Dr. Shamrock,

Thank you for submitting your manuscript to PLOS ONE. After careful consideration, we feel that it has merit but does not fully meet PLOS ONE’s publication criteria as it currently stands. Therefore, we invite you to submit a revised version of the manuscript that addresses the points raised during the review process.

Please note that we have only been able to secure a single reviewer to assess your manuscript. We are issuing a decision on your manuscript at this point to prevent further delays in the evaluation of your manuscript. Please be aware that the editor who handles your revised manuscript might find it necessary to invite additional reviewers to assess this work once the revised manuscript is submitted. However, we will aim to proceed on the basis of this single review if possible. 

We look forward to receiving your revised manuscript.

Kind regards,

Vanessa Carels

Staff Editor

PLOS ONE

Journal Requirements:

3. During your revisions, please confirm whether the wording in the title is correct and update it in the manuscript file and online submission information if needed. Specifically, the title says - Qualitative description of interpersonal HIV stigma and motivations for HIV testing among gays, bisexuals, and men who have men in Ghana's slums- BSGH-005. I believe that it should include the phrase Men who have sex with men.

Reviewers' comments:

Reviewer's Responses to Questions

**Comments to the Author**

1. Is the manuscript technically sound, and do the data support the conclusions?

Reviewer #1: Partly

2. Has the statistical analysis been performed appropriately and rigorously? 

Reviewer #1: N/A

3. Have the authors made all data underlying the findings in their manuscript fully available?

Reviewer #1: Yes

4. Is the manuscript presented in an intelligible fashion and written in standard English?

Reviewer #1: Yes

5. Review Comments to the Author

Reviewer #1: General Observations

This manuscript is about HIV stigma and factors influencing HIV testing among GBMSM in slums of Accra, Ghana. The study employed qualitative data collection methods and reports several findings. On HIV stigma, the study reports that people stay away of avoids GBMSM who are living with HIV and that HIV testing is feared in this sub-population. Factors influencing uptake of HIV testing among GBMSM include perceived vulnerability to HIV, Sexual health decision making, and positive messages about HIV testing. The paper is novel as it addressed literature gaps pertaining to our understanding about the interface between GBMSM and HIV in urban slums. The

“While the study appears to be sound, most of the sections especially in the methods and results sections appear to be underdeveloped. For the methods section, I advise the authors to use the COnsolidated criteria for REporting Qualitative research (COREQ) checklist. For the results section, I suggest authors to provide a discussion or interpretation of the quotes that they have provided. For the Results section, some findings are presented in two or three sentences which suggest that the analysis process was incomplete. I advise the authors to also provide a description or interpretation of the quotes. In some case, language is unclear, making it difficult to follow. I advise the authors review the manuscript to improve the flow and readability of the text. I have provided specific comments below.

Specific Observations

Introduction

Page 3, Paragraph 1, Sentence 2: Kindly specify the context in which these pragmatic efforts occur e.g., despite pragmatic efforts in SSA... or in Ghana....

Page 3, Paragraph 2, Sentence 1: Please indicate '... compared to that of the general population ... '

Page 3, Paragraph 2, Sentence 2: Please avoid repeating the same word in a sentence. Here, GBMSM has already been used. please use the following phrase '... in this group...'

Page 3, Paragraph 2, Sentence 2: Here, only individual, and environmental barriers have been mentioned. However, in subsequent sentences, there is personal, environmental, and interpersonal barriers. Kindly include interpersonal barriers here.

Page 3, Paragraph 2, Sentence 3: Is 'individual level' the same as 'personal level'? Please be consistent in the use of the words.

Page 3, Paragraph 3, Sentence 1: Kindly provide references for these limited studies.

Page 3, Paragraph 3, Sentence 2: This sentence is not coherent with the preceding sentence. There is a disconnect between the first and the second sentences. Please revisit the sentence.

Page 3, Paragraph 3, Sentence 5: Please provide the year when these results were published. For example, in Kenya in 2024, HIV rates among slum...

Page 4, Paragraph 2, Sentence 1: Reference is needed here.

Page 4, Paragraph 2, Sentence 2: This sentence will sound better if written: 'Communities with increased HIV prevalence and risk also have poorly resourced infrastructure and health facilities which contributes to poor health outcomes’.

Page 4, Paragraph 2, Sentence 3: Delete ' where GBMSM were sampled for the study'. This information should be provided in the methodology section.

Page 4, Paragraph 3, Sentence 1: It is important to clarify the type of stigma being referred to here. Is this stigma broadly, self-stigma or other forms of stigma.

Page 4, Paragraph 3, Sentence 3: Excellent to have includes a theoretical framework.

Page 4, Paragraph 3, Sentence 4: Can the theory also explain the HIV stigma dimension that also underpin this study? The theory should be used for all dimension of your study.

Page 4, Paragraph 3, Sentence 6: I think that these dimensions should be part of the discussion section. In the introduction, just describe what the theory is and what are its components. A description of each component is also important in the introductions section.

Methodology

Start this section with the study design. Kindly use the COREQ checklist to guide the content and structure of your methods section.

Page 5, Paragraph 2, Sentence 2: If you are talking about partner organisations, it would be necessary to also mention the main organisation that implemented this work within the methods section.

Page 5, Paragraph 2, Sentence 4: This sentence will read better if it begins like this.

Although we originally intended to purposively include 19 participants in this study, we reached saturation of information after the eighth interview.

Page 5, Paragraph 2, Sentence 4: How did you determine that saturation of information has been attained? Kindly provide examples of elements that you saw in the data to be confident that you reached information saturation.

Page 5, Paragraph 2, Sentence 3: Full stop is needed at the end of the sentence.

Page 5, Paragraph 3, Sentence 2: Kindly use past tense

Page 6, Paragraph 2, Sentence 3: Provide some background information on the qualifications of research assistants to do this work with this group of study participants. If these research assistants are part of the authors, kindly use their initials. E.g., Experienced graduate researchers (GRA and EYZ) conducted the interviews for this study.

Page 6, Paragraph 2, Sentence 5: Write interviews instead of conversations.

Page 6, Paragraph 3, Sentence 1: Please provide the number of days that this training was done and whether, protocol training was part of this training.

Page 7, Paragraph 1, Sentence 1: How was data processing done. Transcription and translation considering that some participants were interviewed in Twi.

Page 7, Paragraph 1, Sentence 1: Are these crucial points or codes?

Results

I would suggest that the authors should use the word 'Findings' instead of 'Results' which is mostly used in quantitative studies.

Page 7, Paragraph 3, Sentence 3: It would be good to include age range of the participant and their current occupation.

Page 7, Paragraph 3, Sentence 2: This sentence should be phrased 'Five participants accomplished tertiary education, six concluded senior high school while one didn't complete Junior High school education.

Page 8, Paragraph 1, Sentence 1: These groups the way this statement is written suggest that they were generated deductively based on researcher’s priori thoughts and not inductively. Kindly confirm if this observation is accurate and include a statement that these categories were deductively generated.

Page 8, Paragraph 1, Sentence 1: Are the main groups themes? Please check the language used to be consistent with language used in qualitative research methodologies.

Page 8, Paragraph 1, Sentence 2: Are the sub-categories sub-themes?

Page 8, Paragraph 1, Sentence 2: Are the sub-categories sub-themes?

Page 8, Paragraph 2, Sentence 1: This sentence will read better if written: Our findings show that study participants feared HIV broadly and avoided being associated with PLHIV in their community. They also lacked proper information about how HIV is transmitted from one person to another.

Page 8, Paragraph 3, Sentence 1: I think that this section is underdeveloped. It needs more information. Include information about why they fear HIV positive test results. In the second quote, the participant mentions some elements of this fear. People usually are afraid of being bonded to lifetime treatment. There may be issues of HIV stigma among other factors.

Page 8, Paragraph 3, Sentence 2: This sentence is not clear and does not link well with previous sentences.

Page 9, Quotes: Some analysis or discussion of what these quotes are portraying is important. For example: The first quote demonstrates how lack of knowledge about HIV transmission precipitates fear and subsequently HIV stigma.

Page 9, Quote 2: I have noted that the authors only use the word 'GBMSM participant' for all quotes. I think that it will be good to add more information such as participant ID number and age so that we are sure that these quotes came from several participants and not just a few of them.

Page 9, Subtitle - Sexual Health Decision Making 1: I suggest that you should revise the subtitle: Knowing ones HIV status

Page 10, Quote 1: The word 'if' should begin with a capital letter.

Discussion section

Page 11, Paragraph 1; Sentence 1: This sentence is not clear. Kindly revise it.

Page 11, Paragraph 1, Sentence 4: This sentence should include the word vulnerability: 'The need to test was driven by factors such as perceived vulnerability...

Page 11, Paragraph 1, Last Sentence: The sentence is missing something. Do you mean that these findings highlight the need for interventions to ...

Page 11, Paragraph 2, Sentence 2: A phrase '...among some participants...' should be either among individuals or among people or among GBMSM.

Page 11; Paragraph 2, Sentence 4: I think that you should delete 'Such interventions remain essential considering that the participants did not only have negative misconceptions about HIV.' A revised sentence should read.

The present study findings of participants projecting stigma towards their peers living with HIV by avoiding them, refusing them sex, and labelling them as sexually promiscuous was also reflected in our previous findings among other GBMSM.

Page 12, Paragraph 1, Sentence 1: I am not clear about a phrase '...opportunities for leveraging motivated GBMSM...'

Page 12, Paragraph 1, Sentence 2: A phrase '...encouraged them to test to protect themselves and their partners.' is not clear. HIV testing does not directly provide protection from HIV infection, but knowledge of HIV status allows individuals to embrace HIV prevention or initiate treatment which also serves as prevention. I suggest that this sentence should end with '...encouraged them to test.'

Page 12, Paragraph 2; Sentence 3: please delete 'also’.

Page 12, Paragraph 2; Sentence 4: delete the word 'to’.

Page 12, Paragraph 2; Sentence 5: delete the word 'also’.

Page 12, Paragraph 2; Sentence 5: I am not sure about the meaning of '...health sexual health behaviour...'

Page 12, Paragraph 3; Sentence 1: I suggest that you should delete a phrase 'To emphasize further...' in this sentence or mention the point that is being emphasised.

Start this sentence as follows: Providing information on how HIV is transmitted...'

Page 13, Paragraph 1; Sentence 2: delete the word 'the'.

Page 13, Paragraph 2; Sentence 1: End this sentence as follows: '...when interpreting these study findings.

Page 12, Paragraph 2; Sentence 2: I think that qualitative designs are important for their depth within the context where the study has been done and not their generalisability or applicability in contexts elsewhere. I think that this limitation does not apply to this study.

Page 12, Paragraph 2; Sentence 4: This also is not a methodological limitation for qualitative studies.

6. PLOS authors have the option to publish the peer review history of their article (what does this mean?). If published, this will include your full peer review and any attached files.

Reviewer #1: No

---

## [Author Response · Author response to Decision Letter 0]

16 Apr 2024

All responses to the reviewer has been attached to this submission in the file "response to reviewer".

---

## [Editor Report · Decision Letter 1]

15 May 2024

PONE-D-23-23534R1Qualitative description of interpersonal HIV stigma and motivations for HIV testing among gays, bisexuals, and men who have men in Ghana's slums- BSGH-005PLOS ONE

Dear Dr. Shamrock,

Thank you for submitting your manuscript to PLOS ONE. After careful consideration, we feel that it has merit but does not fully meet PLOS ONE’s publication criteria as it currently stands. Therefore, we invite you to submit a revised version of the manuscript that addresses the points raised during the review process. We observed that a clean version of the manuscript did not contain the changes made in the tracked version. This suggest that the original version of the clean manuscript was resubmitted instead of a revised manuscript. In the next revision, please make sure that a clean version of the manuscript contain all changes made in the tracked version. Based on changes made in the tracked manuscript, we suggest additional changes to further improve the manuscript. 

We look forward to receiving your revised manuscript.

Kind regards,

Moses Kelly Kumwenda, BEd, MPhil, PhD

Guest Editor

PLOS ONE

Journal Requirements:

Additional Editor Comments:

I would like to thank the authors for working on the manuscript and making appropriate revisions. I have however noted a few issues that needs to be addressed in the tracked version of the manuscript. Based on changes made in the tracked manuscript, I have made additional suggestions to further improve the manuscript. These have been outlined below.

Methods section

Sampling and Recruitment Procedure Section

Revise the sentence: ‘Although we originally intended to purposively include 19 participants in this study, we reached saturation of information after the eight interview.’ The sentence should mention ‘eight interviews’ and not ‘eight interview’.

Inclusion criteria section

Please revise this sentence: ‘Additionally, the individual must identify as a cisgender man who self-identifies as gay, bisexual, or pansexual or engage in sexual intercourse with another cis-gender man for reasons other than sexual orientation’ to 'Additionally, the individual identity had to be a cisgender man who self-identifies as gay, bisexual, or pansexual or engage in sexual intercourse with another cis-gender man for reasons other than sexual orientation.

Data Collection Procedure

Procedure

Revise this sentence: ‘Research assistants (EYZ, OWS) engaged in data collection have had prior and extensive research training from working with these populations in the past’ to ‘Research assistants (EYZ, OWS) engaged in data collection had prior and extensive research training from working with these populations in the past.’

Revise this sentence: ‘We have also utilized the expertise of these researchers in our past studies, with similar populations, hence, the familiarity and experience engaging our participants’ to ‘We also utilized the expertise of these researchers in our past studies, with similar populations, hence, the familiarity and experience engaging our participants.’

Ethical Considerations

Revise the word 'Participant' to 'participant' with a lower case.

Results

Description of findings

Rewrite the word 'Motivation' to 'motivation' in a sentence: ‘Findings from the data collected and analyzed revealed two categories; 1) HIV stigma and 2) Motivations for HIV testing.’

HIV Stigma: Avoidance of GBMSM living with HIV

Please revise this sentence: ‘Participant K, mentioned they were scared to be near someone with HIV because they worried, they might get infected’ to ' Participant K, mentioned they were scared to be near someone because of fear of a potential HIV infection’.

Please revise this sentence: ‘Some participants knew a lot about how HIV spreads and were ok with being around people who have HIV’ to ‘Some participants were well informed about how HIV is transmitted and were comfortable with being around people who have HIV.’

Discussion

Please revise this sentence: ‘We employ the SDT theory in the context of this study to highlight the psychological needs for autonomy, competence, and relatedness for GBMSM in HIV testing, or how this affects the motivation for testing.’ To ‘'The SDT theory was useful in the context of this study to highlight the psychological needs for autonomy, competence, and relatedness for GBMSM in HIV testing, or how this affects the motivation for testing.'

Please revise this sentence: ‘The theory also throws more light on HIV stigma and how this impacts autonomy as fears of negative stereotypes impede testing.’ To ‘The theory also improved our understanding of HIV stigma and how it impacts autonomy considering that the fear of negative stereotypes impedes uptake of testing.’

Please revise this sentence: ‘The need to test was driven by factors such as perceived HIV vulnerability, need to know one’s HIV status for sexual health reasons, and positive messaging about HIV-informed motivations for HIV Testing. Revise the word 'Testing' to 'testing’.

Reference is needed to support a sentence ‘The present study findings of participants projecting stigma towards their peers living with HIV by avoiding them, refusing them sex, and labelling them as sexually promiscuous was also reflected in our previous findings among other GBMSM.’ If the previous study has not been published, it is important to indicate '...was also reflected in our previous unpublished findings …'

References

The authors should be consistent with their referencing. They should use Plos One guidance on how to cite sources. For example, reference number 5 also has authors qualifications but other do not.

---

## [Author Response · Author response to Decision Letter 1]

16 May 2024

Methods

1. Eight interview-was changed to eight interviews. 

2. Additionally, the individual must identify as a cisgender man who self-identifies as gay, bisexual, or pansexual or engage in sexual intercourse with another cis-gender man for reasons other than sexual orientation- Has been revised to Additionally, the individual identity had to be a cisgender man who self-identifies as gay, bisexual, or pansexual or engage in sexual intercourse with another cis-gender man for reasons other than sexual orientation.

3. Research assistants (EYZ, OWS) engaged in data collection have had prior and extensive research training from working with these populations in the past- Has been revised to Research assistants (EYZ, OWS) engaged in data collection had prior and extensive research training from working with these populations in the past.

4. We have also utilized the expertise of these researchers in our past studies, with similar populations, hence, the familiarity and experience engaging our participants- Has been revised to We also utilized the expertise of these researchers in our past studies, with similar populations, hence, the familiarity and experience engaging our participants.

5. “Participants”- has been changed to “participants”.

Results

6. “Motivation”- has been changed to “motivation”.

7. Under HIV stigma, we identified two subcategories, avoidance of GBMSM living with HIV, and fear of testing positive for HIV- has been changed to Under HIV stigma: avoidance of GBMSM living with HIV, and fear of testing positive for HIV

8. Under the category motivations for HIV testing, we identified three subcategories: HIV vulnerability, sexual health decision-making, and knowing one’s HIV status - has been changed to Under the category motivations for HIV testing: HIV vulnerability, sexual health decision-making, and knowing one’s HIV status.

9. Participant K, mentioned they were scared to be near someone with HIV because they worried, they might get infected - has been changed to Participant K, mentioned they were scared to be near someone because of fear of a potential HIV infection.

10. Some participants knew a lot about how HIV spreads and were ok with being around people who have HIV - has been changed to Some participants were well informed about how HIV is transmitted and were comfortable with being around people who have HIV.

Discussion

11. We employ the SDT theory in the context of this study to highlight the psychological needs for autonomy, competence, and relatedness for GBMSM in HIV testing, or how this affects the motivation for testing. - has been changed to The SDT theory was useful in the context of this study to highlight the psychological needs for autonomy, competence, and relatedness for GBMSM in HIV testing, or how this affects the motivation for testing.'

12. The theory also throws more light on HIV stigma and how this impacts autonomy as fears of negative stereotypes impede testing. - has been changed to The theory also improved our understanding of HIV stigma and how it impacts autonomy considering that the fear of negative stereotypes impedes uptake of testing.

13. Testing - has been changed to testing

14. Reference has been provided for “The present study findings of participants projecting stigma towards their peers living with HIV by avoiding them, refusing them sex, and labelling them as sexually promiscuous was also reflected in our previous findings among other GBMSM in Ghana.” 

References

15. Number (5) reference has been updated to Price MA, Rida W, Mwangome M, Mutua G, Middelkoop K, Roux S, et al. Identifying at-risk populations in kenya and south africa: HIV incidence in cohorts of menwho report sex with men, sex workers, and youth. J Acquir Immune Defic Syndr (1988). 2012;59(2). doi:10.1097/QAI.0b013e31823d8693

16. Added reference: 31. Dakpui HD, Shamrock OW, Aidoo-Frimpong G, Zigah EY, Agbemedu GR, Ahmed A, et al. A qualitative description of HIV testing and healthcare experiences among trans women in Ghanaian urban slums BSGH-011. 2024. doi:https://doi.org/10.21203/rs.3.rs-4390892/v1

17. Added reference: 6. Abu-Ba’are GR, Torpey K, Nelson L, Conserve D, Jeon S, McMahon J, et al. Adaptation and feasibility of Many Men Many Voices (3MV), an HIV prevention intervention to reduce intersectional stigma and increase HIVST among YSMM residing in Ghanaian slums–A clustered pre-post pilot trial protocol. 2024. Available from: doi: 10.21203/rs.3.rs-4313437/v1

18. Added reference: 20. Osman Wumpini Shamrock, Henry Delali Dakpui, George Rudolph Kofi Agbemedu, Donte T Boyd, Kharul Islam, Ibrahim Wunpini Mashoud, et al. “I’m scared of the nurses telling other people I am a transwoman”: Disclosure and nondisclosure of gender identity among trans women in Ghana’s urban slums– BSGH010 [Internet]. 2024 Apr [cited 2024 Apr 19]. Available from: https://doi.org/10.21203/rs.3.rs-4243840/v1

19. All other references have been adjusted to fit the journal’s specifications.

---

## [Editor Report · Decision Letter 2]

21 May 2024

Qualitative description of interpersonal HIV stigma and motivations for HIV testing among gays, bisexuals, and men who have men in Ghana's slums- BSGH-005

PONE-D-23-23534R2

Dear Dr. Osman Wumpini Shamrock,

We’re pleased to inform you that your manuscript has been judged scientifically suitable for publication and will be formally accepted for publication once it meets all outstanding technical requirements.

Kind regards,

Moses Kelly Kumwenda, BEd, MPhil, PhD

Guest Editor

PLOS ONE

Additional Editor Comments (optional):

There are several formatting issues that needs addressing.

1. References should be in square brackets [].

2. There should be consistency in placing of citations. In some cases, there is space between the last word and the reference bracket (e.g. ...participants (24,25). In others, there is no space (e.g. ...community stigma among others(12–15).

3. In a sentence 'Sub-Saharan Africa (SSA) carries over two-thirds of the world's burden of HIV, yet, HIV testing and related services remain underutilized due to several factors such as insufficient

knowledge, low-risk perception, and increased stigma. (1,2).' there is a full stop between the word 'stigma' and the reference '(1,2)'. This is different from how most of the references have been presented.

4. In a sentence 'Emerging findings as from 2010 in eastern Africa show that GBMSM in slums have low-risk perception, increased risk behaviors, and low access to HIV testing and prevention services.(21,22).' there is also a full stop between the last word and the reference.

5. Authors should be consistent in their presentation of the participants information for quotes. In some cases, there is a full stop between the quote and participants information {e.g. and told me they were positive. (GBMSM participant E)} in some cases there is no full stop {e.g. '...I've had raw sex with someone, and I don't trust the person (GBMSM participant D)}. In the same manner, other quotes have a full stop at the end of participant information {e.g. (GBMSM participant A).}, while others do not have the full stop {e.g. '...so I will neglect the person. (GBMSM participant K)}

Conclusion should have a separate sub-title 'Conclusion' in bold
---

## [Editor Report · Acceptance letter]

27 May 2024

PONE-D-23-23534R2 

PLOS ONE

Dear Dr. Shamrock, 

I'm pleased to inform you that your manuscript has been deemed suitable for publication in PLOS ONE. Congratulations! Your manuscript is now being handed over to our production team.

Kind regards, 

on behalf of

Dr Moses Kelly Kumwenda 

Guest Editor

PLOS ONE